# Hyperelongate ornamental tail feathers in a new early *Cretaceous enantiornithine* bird

Alexander D. Clark[1,2]*, Jingmai K. O'Connor[2], Xiaoli Wang[3,4]*, Yan Wang[3,4,5], Stephen Pruett-Jones[5], Xiangyu Zhang[3,4], Xing Wang[3,4], Xiaoting Zheng[3,4], Zhonghe Zhou[6,7]

1 Committee on Evolutionary Biology, University of Chicago, Chicago, Illinois, United States of America, 2 Negaunee Integrative Resource Center, Field Museum of Natural History, S Dusable Lake Shore Drive, Chicago, Illinois, United States of America, 3 School of life Sciences, Lingyi University, Lingyi, China, 4 Shandong Tianyu Museum of Nature, Pingyi, China, 5 Department of Ecology and Evolution, University of Chicago, Chicago, Illinois, United States of America, 6 Key Laboratory of Vertebrate Evolution and Human Origins of the Chinese Academy of Sciences, Institute of Vertebrate Paleontology and Paleoanthropology, Beijing, China, 7 Chinese Academy of Sciences Center for Excellence in Life and Paleoenvironment, Beijing, China

* adclark@uchicago.edu (ADC); wang_7355@163.com (XW)

## Abstract

Bird diversity is reflected in the abundance and variety of extraordinary plumages. Some of these include elongate, ornamental tail feathers that are typically attributed to either intraspecific communication in monomorphic species or sexual selection in sexually-dimorphic ones. Enantiornithines (Aves: Ornithothoraces) were the most diverse group of birds during the Cretaceous. Importantly, some enantiornithine fossils preserve soft tissues, most often in the form of feathers surrounding the body. Unlike any living bird, many enantiornithine specimens lack tail feathers (rectrices) all together, with the tail region consisting entirely of contour feathers. However, when present, enantiornithine rectrices typically consist of a pair of elongate, ornamental feathers with unusually wide rachises, referred to as rachis-dominated feathers. Here we describe *Plumadraco bankoorum* gen. et sp. nov., a new bohaiornithid enantiornithine with a pair of exceptionally long rectrices. These tail feathers measure twice the individual's body length, ending in proportionally small pennaceous rackets, thus adding to the growing diversity of these unusual feathers. The fine preservation of these tail feathers, in comparison to other enantiornithine rectrices, reveals previously unrecognized structural variation that hints at their potential function in courtship displays. Although ornamental feathers in enantiornithines are widely considered sexually dimorphic, determining the selection pressures that shaped them is difficult due primarily to limited soft tissue data. Enantiornithine rectrices are likely the result of an interplay between both sexual and naturally selective pressures, similar to the processes which produce analogous structures in birds today.

**Data availability statement:** All relevant data are within the paper and its Supporting Information files.

**Funding:** This research was in part funded by the Taishan Scholar Foundation of Shandong Province (Ts20190954) and the National Natural Science Foundation of China (NSFC), grant numbers (42288201, 42572027).

**Competing interests:** The authors have declared that no competing interests exist.

## Introduction

Enantiornithines were the most speciose clade of Mesozoic birds, with over 100 genera described to date and specimens recovered from all continents except Antarctica [1-3]. Spectacular preservation of enantiornithine specimens from the Early Cretaceous Jehol Biota (130−120 Ma) in northeastern China often includes soft tissue structures, most commonly in the form of feathers [4-7]. Body (contour) feathers, followed by remiges, are the most prevalent, with those of the tail (rectrices) being comparatively rare. The majority of enantiornithine specimens preserving soft tissues lack tail feathers all together, and instead only have body contour feathers covering the tail region – a condition absent among all extant neornithines.

When present, enantiornithine rectrices typically consisted of a single pair of elongate ornamental (non-aerodynamically assisting) feathers with proportionately wide rachi throughout majority of their lengths, referred to as rachis-dominated feathers (RDFs) [8,9]. RDFs, present in both enantiornithines and confuciusornithiforms, are considered modified pennaceous feathers representing an extinct feather morphotype [8]. Three-dimensionally preserved specimens in amber reveal that instead of the pith-filled tube-like rachis of neornithines, RDFs have a ventrally-open "C"-shaped cross-section, indicating they developed differently and had disparate tensile properties compared to similarly elongate rectrices present in extant birds [10,8]. Specimens with RDFs have also been recovered in the Lower Cretaceous Xiagou Formation in northwestern China, the Lower Cretaceous Crato Formation of Brazil, and Upper Cretaceous Kachin amber from Myanmar, indicating these feathers were widespread within enantiornithines and persisted for at least 30 million years [10,11,8]. Most specimens preserving these feathers, and thus the vast majority of data regarding their diversity, are from the Jehol deposits [7], where they have been documented in bohaiornithids [12,13], pengornithids [4,9], and tentatively in the longipterygid *Shanweiniao* [14,6], as well as in other taxa whose familial relationships remain unclear at this time (e.g., *Protopteryx, Dapingfangornis, Feitianius*) [15,16,5,17].

The Bohaiornithidae is a diverse group of enantiornithines from the Jehol Biota, with members ranging in size between ~50–200 g using humeral length as a proxy for estimating body mass [18,19]. Bohaiornithids are readily characterized by their robust teeth, proportionally long minor metacarpal, and pedal digits with comparatively large claws [12,20-22]. Though bohaiornithids have previously been documented with poorly preserved, incomplete RDFs (e.g., *Bohaiornis*, CUGB P1202, *Neobohaiornis*), no specimen thus far preserves a complete rectrix [23,13,22]. Here we describe a new specimen of bohaiornithid enantiornithine from Jehol deposits that preserves a pair of hyperelongate RDFs measuring twice the individual's body length – the proportionally longest tail feathers recorded in any enantiornithine. Although much of the skeleton is poorly preserved, the feathers are among the best-preserved exemplars of RDFs recovered thus far, allowing for detailed examination of their structure. We use cladistic analysis to explore the relationship between this new taxon and other previously described enantiornithines and discuss the role of RDFs in enantiornithine biology.

Institutional Abbreviations: BMNHC Beijing Museum of Natural History, Beijing, China; CUGB – China University of Geology, Beijing, China; FMNH – Field Museum of Natural History, Chicago, United States of America; GSGM – Gansu Geological Museum, Lanzhou, China; IVPP – Institute of Vertebrate Paleontology and Paleoanthropology, Beijing, China; STM – Shandong Tianyu Museum of Nature, Pingyi, China.

## Materials and methods

STM11−4 is housed at the Shandong Tianyu Museum of Nature (Pingyi, China). All necessary permits were obtained for the described study, which complied with all relevant regulations. Photographs of the specimen were taken with a Canon 5DM3 and imported into Adobe Photoshop version 25.2.0. Anatomical nomenclature primarily follows Baumel and Witmer (1993) [24] using the English equivalents of the Latin terminology. Terminology of rectrix structures draws from Carroll et al., (2019), O'Connor et al., (2012), and Foth (2020) [10,8,25]. Measurements of the specimen were taken first-hand to the nearest 0.01 mm using digital calipers (OriginCal1P54). Additional measurements were taken in ImageJ [26], utilizing the scale measuring tool calibrated to in-photo scales. Principal component analyses (PCA) of x-ray fluorescence (XRF) data (ThermoFisher Scientific Portable XRF Analyzer Niton XL3t 950 Instrument S/N 93368) was used to geochemically support the preservation of body, wing, and rectrices and explore their elemental composition. PCA of the XRF data was performed using the program MURRAP created by the University of Missouri's archaeometry laboratory [27].

Phylogenetic Analysis. To explore the phylogenetic position of STM11−4, it was added to a modified version of the Shen et al., (2024) [13] matrix. Like these authors, we removed *Elsornis* and *Eoenantiornis* due to their unstable position in preliminary analyses. The final matrix, consisting of 55 taxa with scores across 212 characters, was analyzed using the parsimony-based algorithm in TNT [28] (see supplemental information).

## Nomenclatural acts

The electronic edition of this article conforms to the requirements of the amended International Code of Zoological Nomenclature, and hence the new names contained herein are available under that Code from the electronic edition of this article. This published work and the nomenclatural acts it contains have been registered in ZooBank, the online registration system for the ICZN. The ZooBank LSIDs (Life Science Identifiers) can be resolved and the associated information viewed through any standard web browser by appending the LSID to the prefix ""http://zoobank.org/"."

The LSID for this publication is: urn:lsid:zoobank.org:pub:D64D623C-8E0A-4592-8C6E-B9881C3F104F. The LSID for the new genus is: urn:lsid:zoobank.org:act:6FE850E5-CDC1–4830-A2B2-14501BB3BB43. The LSID for the new species is: urn:lsid:zoobank.org:act:73483921-C9B2-4830–8720-E13823C71379. The electronic edition of this work was published in a journal with an ISSN, and has been archived and is available from the following digital repositories: PubMed Central, LOCKSS.

## Systematic paleontology

Aves Linnaeus, 1758

 Pygostylia Chiappe et al., 2002 [1]

 Ornithothoraces Chiappe, 1995 [29]

 Enantiornithes Walker, 1981

 *Plumadraco bankoorum* gen. et sp. nov. (Fig 1, Table 1)

**Etymology.** "Pluma", Latin for feather, and "draco", Latin for dragon. In the theme of avian biology and evolution, the specific name, "bankoorum", honors Winston E. and Paul C. Banko. Together, their momentous life-long efforts have significantly contributed to our understanding of avian biology and conservation, particularly across the Hawaiian archipelago. *Plumadraco bankoorum*, the Banko's feather dragon.

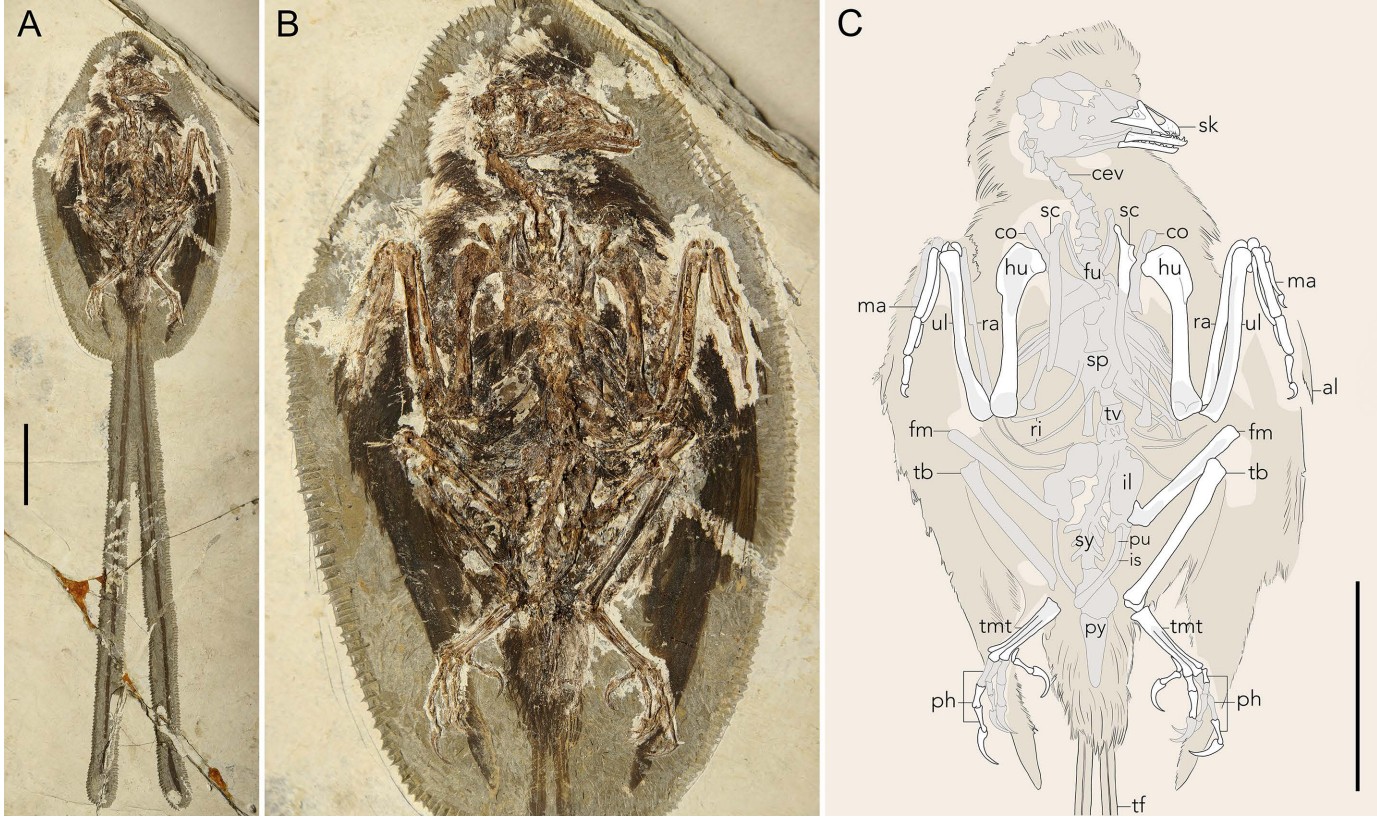

**Fig 1. Specimen STM11−4 (*Plumadraco bankoorum*). (A)** The holotype specimen of *Plumadraco*, **(B)** a closer view of the body and, **(C)** a line draw-ing of the same portion of the body shown in **B**. Well preserved bones are colored white, crushed or poorly preserved bones (or portions of bones) are grey, and preserved soft tissues are brown. The extent of the feather traces are denoted by the thin black outline around the body. Potential outline or crural feather present along the cranial face of the tibia. Abbreviations: al, alula; cev, cervical vertebrae; co, coracoid; fm, femur; fu, furcula; hu, humerus; il, ilium; is, ischium; ma, manus; ph, phalanges (of the peds); pu, pubis; py, pygostyle; ri, rib(s); ra, radius; sc, scapula; sk, skull; sp, sternal plate; sy, synsacrum; tb, tibia; tmt, tarsometatarsus; tf, tail feathers; tv, thoracic vertebrae; ul, ulna. Scale bar (A, C) equals 50 mm.

**Holotype.** STM11−4 is a complete, articulated specimen preserved in a single slab primarily in dorsal aspect with feathers preserved around the head, body, wings, and tail.

**Locality and horizon.** Near Xiaotaizi Village, Jianchang County, Liaoning Province, Jiufotang Formation, 121 Ma (Lower Aptian) [2,30].

**Diagnosis.** Mid-sized [112–144 g, similar to some extant turdids (e.g., *Cochoa*, *Turdus*) and meliphagids (e.g., *Anthochaera*) [18,19,31] enantiornithine (ventral margin of the furcula wider than dorsal margin; acrocoracoid, glenoid, and scapular cotyla omal-sternally aligned; minor metacarpal extending distally farther than the major metacarpal; metatarsal IV thinner than metatarsals II and III with the trochlea reduced to a single condyle; and a J-shaped metatarsal I), belonging to the family Bohaiornithidae (basally robust, apically tapered dentition; unforked dentary- surangular articulation; proportional width of the coracoid's sternal margin; caudolateral projection of the sternal plate's lateral trabeculae; well-developed abruptly terminating deltopectoral crest; robust pedal unguals), with the unique combination of the following features: corpus of the premaxillae dorsoventrally deeper than the dentaries; rostral ~80% of the dentary with parallel dorsal and ventral margins; tip of dentary is rostrodorsally tapered; at least nine sacral vertebrae; caudally-oriented lateral trabeculae of the sternum with asymmetrical, fan-shaped distal expansions; phalanx I of the manual digit craniocaudally thin; weakly curved pedal unguals; RDFs approximately twice body length.

**Table 1. Skeletal measurements of specimen STM11−4 (*Plumadraco bankoorum*) in mm. Asterisks represent approximations due to poor preservation.**

| (Measurements in mm) | Left | Right |
|---|---|---|
| Humerus Length | 42.1 | 41.69 |
| Humerus Proximal width | 9.75 | 10.1 |
| Humerus Distal width | 7.87 | 7.79 |
| Radius Length | 38.95 | 39.64 |
| Radius midpoint thickness | 2.14 | 2.21 |
| Ulna Length | 41.09 | 41.81 |
| Ulna midpoint thickness | 3.14 | 3.3 |
| Alular Metacarpal | – | 4 |
| Alular phalanx I | – | 8.59 |
| Alular phalanx II | 4.63 | 4.33 |
| Major Metacarpal | 17.31 | 16.94 |
| Major digit phalanx I | 9.28 | 9.01 |
| Major digit phalanx II | 5.67 | 6.55 |
| Major digit phalanx III | 3.82 | 4.34 |
| Minor Metacarpal | 18.72* | 18.25 |
| Minor digit phalanx I | 4.49 | 4.86 |
| Femur Length | 33.88 | 34.13* |
| Tibia Length | 40.6 | 40.95 |
| TMT Length | 20.11 | 20.35 |
| Pedal Digit I Phalanx I | 5.45 | 5.47 |
| Pedal Digit II Phalanx I | 5.94 | 6 |
| Pedal Digit II Phalanx II | 6.62 | 6.32 |
| Pedal Digit III Phalanx I | 7.23 | – |
| Pedal Digit III Phalanx II | 5.89 | – |
| Pedal Digit III Phalanx III | 5.98 | 5.68 |
| Pedal Digit IV phalanx I | – | 3.5 |
| Pedal Digit IV phalanx II | – | 3.64 |
| Pedal Digit IV phalanx III | – | 3.05 |
| Pedal Digit IV phalanx IV | 4.25 | 4.53 |
| – | – | – |
| Skull | 38.5* | |
| Synsacrum | 31.88* | |
| Pygostyle Length | 20.2* | |

## Results

### Anatomical description

**Skull.** The skull is preserved in right lateral aspect (Fig 2). The premaxillary corpus is dorsoventrally deeper than the dentaries, atypical for enantiornithines with the exception of *Chiappeavis* and possibly *Parabohaiornis* [6,22]. The craniodorsal margin of the premaxilla is broadly convex, similar to *Bohaiornis*, but unlike many enantiornithines in which the rostrum is more sharply tapered [1,5]. The frontal process of the premaxilla is strap-like, and although the distal end is poorly preserved, appears to extend at least to the rostral border of the antorbital fenestra. The ventral margins of the premaxilla and maxilla are straight. Five teeth are preserved in the right upper jaw: three in the premaxilla and

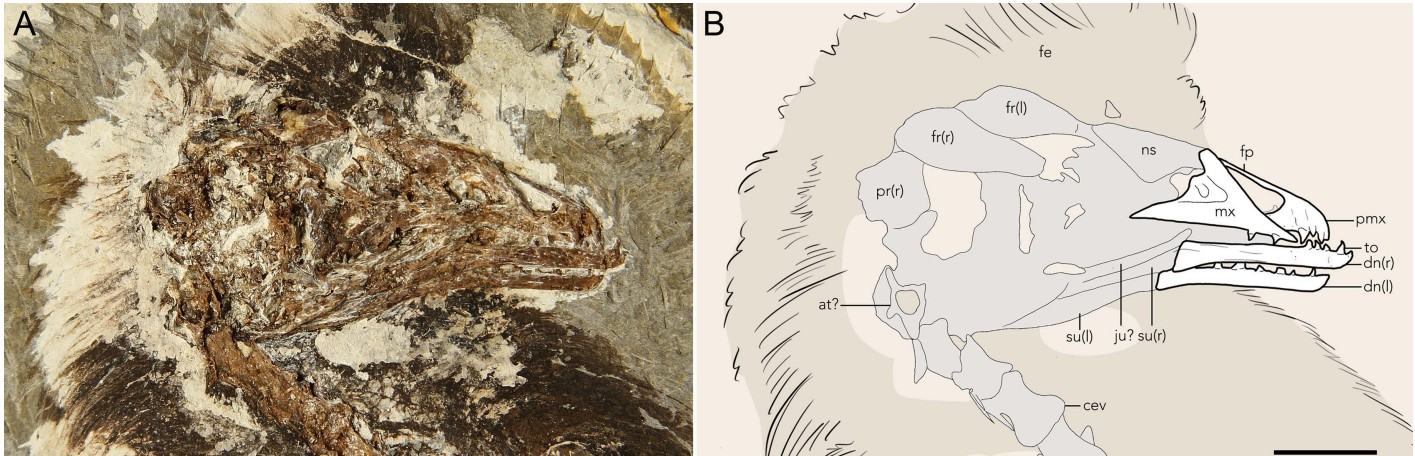

**Fig 2. The skull of STM11−4 (*Plumadraco bankoorum*). (A)** Photo of fossil specimen and **(B)** a corresponding line drawing. Well preserved bones are colored white, crushed or poorly preserved bones (or portions of bones) are grey, and preserved soft tissues are brown. Abbreviations: at, atlas; cev, cervical vertebrae; dn, dentary; fe, feathers; fp, frontal process of the premaxilla; fe, feather; fr, frontal; l, left; mx, maxilla; ns, nasal; pmx, premaxilla; pr, parietal; r, right; su, surangular; to, tooth. Scale bar 10 mm.

two in rostral-most portion of the maxilla. The teeth are nearly uniform in size, apically pinched, and mesially and distally convex, similar to dental morphologies observed in bohaiornithids (e.g., BMNHC-Ph1204) [32,20]. A maxillary foramen, like that in *Longusunguis* [22], appears to be present, though cannot be confirmed due to poor preservation. The right nasal, preserved laterally, expands caudally to form the caudal margin of the external nares at its widest point, caudal to which it tapers towards its articulation with the frontal. The right and left dentaries are preserved in lateral and medial aspects respectively. The rostral end of each dentary tapers to a point that is subtly dorsally-oriented as in bohaiornithids and longipterygids [33-35]. This contrasts with pengornithids which exhibit a more rounded rostral margin of the dentary [1,36]. The right and left dentaries each preserve four and six in situ teeth respectively that appear morphologically similar to those in the premaxilla and maxilla. In both dentaries the dorsal and ventral margins appear parallel for approximately 80% of their lengths, similar to *Bohaiornis* [12]. The dentary subtly expands caudally to meet the surangular, contacting this element through a caudoventrally sloping and unforked articulation, as in other bohaiornithids [37]. The caudoventral margin of the dentary only weakly extends beyond the caudodorsal margin, similar to BMNHC-Ph1204, contrasting with *Bohaiornis* and *Parabohaiornis* in which the caudoventral margin terminates markedly beyond the caudodorsal margin [15,22].

**Axial skeleton.** Eight poorly preserved cervical vertebrae are visible. A ring-like structure at the base of the skull with a subcircular neural canal interpreted as the atlas. The post-atlantal cervicals are longer than they are wide and the prezygapophyses are cranially oriented whereas the postzygapophyses are slightly splayed caudolaterally. The articulated thoracic vertebrae overlap the sternum, but are heavily crushed with the exception of the distal two which preserve lateral excavations typical of enantiornithines. The synsacrum consists of at least of nine fused vertebrae, the caudal five of which exhibit caudolaterally-oriented transverse processes. Like *Neobohaiornis*, *Plumadraco* has two more sacral vertebrae than is typical of enantiornithines [1,13]. The pygostyle is preserved in dorsal aspect, and is triangular with a bluntly tapered distal margin, similar in morphology to other bohaiornithids (e.g., *Parabohaiornis*, *Sulcavis*), but lacking the distal constriction present in many enantiornithines (i.e., longipterygids, *Dapingfangornis*) [20].

**Pectoral girdle.** The ventral margin of the furcula is wider than the dorsal margin such that the rami are dorsolaterally excavated, typical of enantiornithines [1]. The furcular rami are laterally bowed such that the weakly expanded omal tips are oriented craniomedially, contrasting with the craniolaterally orientated rami of other bohaiornithids (e.g.,

*Neobohaiornis*) [13] (Fig 3). The interclavicular angle measures ~34° based on the observable portions of the furcula, which is much narrower than observed in other bohaiornithids (*Bohaiornis* 64.5°, *Sulcavis* 65.5°, *Shangyang* 47°). The left coracoid is in articulation with the sternum, with the proportional width of the sternal margin (to that of the coracoid's total length) being similar to other bohaiornithids (*Plumadraco* 50%, *Parabohaiornis* 50%, *Longusunguis* 52%, BMNHC-Ph1204 52%) [32,22]. The coracoid's elongate neck exceeds the length of the triangular corpus (corpus length ~66% that of the neck). Visible on the right coracoid, the acrocoracoid, glenoid, and scapular cotyla are omal-sternally aligned, typical of enantiornithines [1]. The scapulae are predominately straight, as in most enantiornithines, with bluntly tapered distal margins [1]. The acromion process is long (glenoid length ~77% that of the acromion process) and is bluntly expanded, similar to other bohaiornithids (e.g., *Gretcheniao*, *Parabohaiornis*) [15,22].

The sternum has a rounded cranial margin lacking craniolateral processes. The left lateral trabecula is caudolaterally oriented, similar to bohaiornithids [12,33,22]. The right lateral trabecula is caudally-oriented, likely due to taphonomic displacement. The lateral trabeculae have fan-shaped distal expansions that are weakly asymmetrical with greater medial inflation, but lack the well-developed, sharply-tapered distomedial expansions present in other bohaiornithids [37–39]. As preserved, the xiphoid process terminates beyond the lateral trabecula (Fig. 3). Thoracic ribs and gastralia are preserved disarticulated throughout the pectoral girdle. Two straight uncinate processes with distally rounded margins project at approximately 70° from two caudal thoracic ribs.

**Forelimb.** The humeri are exposed in caudal aspect. The proximal margin of the humerus is 'saddle-shaped', typical of enantiornithines [1] (Fig 3). The humerus is weakly sigmoidal, similar to *Dapingfangornis*, *Eocathayornis*, *Fortunguavis*, and *Parapengornis* [4,5,40,38,41]. This in contrast to the more well-developed sigmoidal morphology present in some other enantiornithines (e.g., *Chiappeavis*, *Misuvavis*, *Sulcavis*) [20,6,42]. The deltopectoral crest is well developed (~ 37% of shaft length) and dorsoventrally deep (~ 45% dorsolateral depth of the proximal humerus). It abruptly truncates distally, similar to some bohaiornithids (e.g., CUGB P1202, BMNHC Ph1204, *Gretcheniao*) [15,32,23], but contrasts with the more

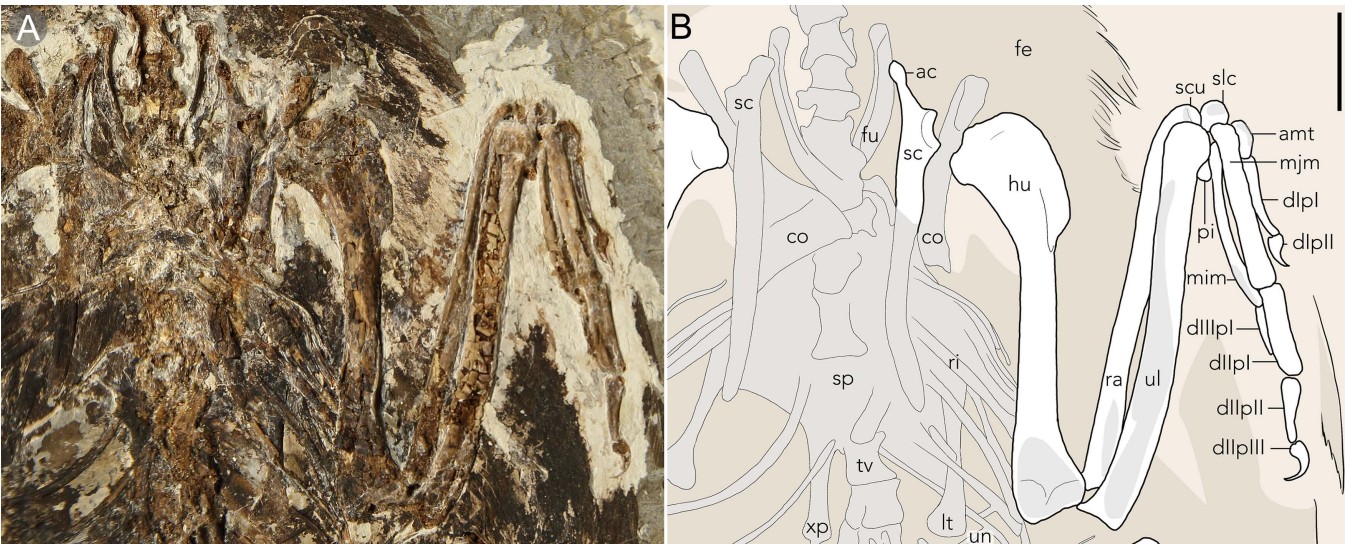

**Fig 3. The upper right forelimb and partial pectoral girdle of STM11−4 (*Plumadraco bankoorum*). (A)** Photo of the fossil specimen's right pectoral girdle and forelimb and **(B)** a corresponding line drawing. Well preserved bones are colored white, crushed or poorly preserved bones (or portions of bones) are grey, and preserved soft tissues are brown. Abbreviations: ac, acromiom process; amt, alular metacarpal; co, coracoid; d, digit; fe, feathers; fu, furcula; hu, humerus; mim, minor metacarpal; mjm; major metacarpal; p, phalanx; rad, radialis; ri, rib(s); lt, lateral trabecula; p, phalanx; pi, pisiform; ra, radius; sc, scapula; scu, scapholunare; slc, semilunate carpal; sp, sternal plate; tv, thoracic vertebrae; ul, ulna; up; uncinate process; xi, xiphoid process. Scale bar equals 10 mm.

gradual termination present in *Neobohaiornis* and *Longusunguis* [13,22]. Caudally, the capital incisure separates the humeral head from the ventral tubercle.

Both ulnae and radii are preserved in caudal aspect. The proximal half of the ulna is bowed, typical of early birds. As in most enantiornithines, the olecranon process is weakly developed. The ventral and dorsal condyles are convex. The carpal tubercle appears to project lateroventrally. The radii are approximately 95% the length of the ulnae, and are straight for most of their length, except for the distal ends which are weakly ventrally deflected. The midshaft width of the right radius measures approximately 67% that of the ulna. In the wrist, the pisiform is badly crushed, but the exposed surface appears convex. The scapholunare appears loosely rectangular in shape but is moderately crushed. The semilunate carpal is not fused to the major and minor metacarpals, suggesting osteological immaturity.

Both manus are exposed in dorsal view. The alular metacarpal is rectangular with a weakly convex cranial margin. The major metacarpal is craniocaudally wider than either the alular or minor metacarpal (alular metacarpal 64%, and minor metacarpal 74% the width of the major metacarpal). As in bohaiornithids (e.g., BMNHC-Ph1204, *Parabohaiornis*, *Neobohaiornis*) [32,22] and longipterygids (e.g., *Longipteryx*, *Rapaxavis*) [43,35], the minor metacarpal extends beyond the major metacarpal to a greater degree relative to other enantiornithines.

Alular phalanx I is gently bowed along its length so that the cranial margin is concave and the caudal margin is convex as in the bohaiornithids *Sulcavis* and *Gretcheniao* [15,20]. It measures approximately 51% the length of the major metacarpal, longer than most enantiornithines (e.g., *Sulcavis* 46%, *Parabohaiornis* 44%, *Gretcheniao* 45%, *Neobohaiornis* 46%) [15,13,20,22]. The ungual phalanx is strongly curved and poorly preserves a keratinous sheath that would further increase its curvature. Major digit phalanx I is rectangular and subequal in mediolateral thickness to the major metacarpal. Major digit phalanx II distally tapers as in bohaiornithids (e.g., BMNHC-PH1204, *Bohaiornis*) [44,32]. The ungual of the major digit is slightly smaller than that of the alular digit. Phalanx I of the minor digit is much thinner than the minor metacarpal and the major and alular digit phalanges. The bone is approximately 56% the mediolateral width and 57% the proximodistal length of alular phalanx I. The first phalanx of the minor digit is long and narrow, similar to *Shangyang* and contrasting with the lower aspect ratios of most enantiornithines (e.g., *Sulcavis*, *Parabohaiornis, Neobohaiornis, Longusunguis*) [20,13,22].

**Pelvic girdle.** The dorsally exposed pelvis is heavily crushed. The ilia are separated medially. The right ilium is exposed medially and the left is exposed laterally. The dorsal margin appears straight. The preacetabular wings of the ilia are craniolaterally inflated. The ventral margin cranial to the pubic pedicel is concave. The pubis and ischium are bowed such that the medial margins are concave and the lateral margins are convex. The preserved portions of the pubes approach each other suggesting they contacted distally.

**Hindlimb.** The femora are straight, similar to other bohaiornithids (e.g., *Parabohaiornis*, *Gretcheniao*), *Imparavis,* and some Late Cretaceous enantiornithines (e.g., PVL 4038) [15,1,22,45]. The femoral neck projects medially, but the femoral head not preserved. Visible on the left, the trochanteric crest has a prominent cranial projection. The medial and lateral condyles are rounded and terminate distally at the same level. The femur measures approximately 88% the length of the tibia, similar to *Sulcavis* and *Yuanchuavis* [20,9]. Distally, the astragalus, with its well-developed ascending process, is not fused to the tibia, another indication of osteological immaturity. The fibular crest is well developed, extending for nearly a quarter of the length of the tibia.

Metatarsals II-IV are fused proximally to the distal tarsals forming a tarsometatarsus (Fig 4). Both tarsometatarsi are preserved in plantar aspect, with poorly preserved epiphyses. The tarsometatarsus is waisted, being mediolaterally narrowest at its midpoint, as in other bohaiornithids [22]. Metatarsal I, preserved on both peds, is J-shaped, with a distal articular surface perpendicular to the lateral articular surface with metatarsal II. At their midpoints, metatarsals II and III are subequal in width, whereas metatarsal IV is mediolaterally compressed, typical of enantiornithines (~ 49% the mediolateral thickness of metatarsals II and III at their midpoint). The trochlea of metatarsal IV terminates first, followed by II, and then III, as in most enantiornithines [44,40]. The trochlea of metatarsal II is ~17% mediolaterally wider than that of IV, which is reduced to a single condyle as in other enantiornithines.

 

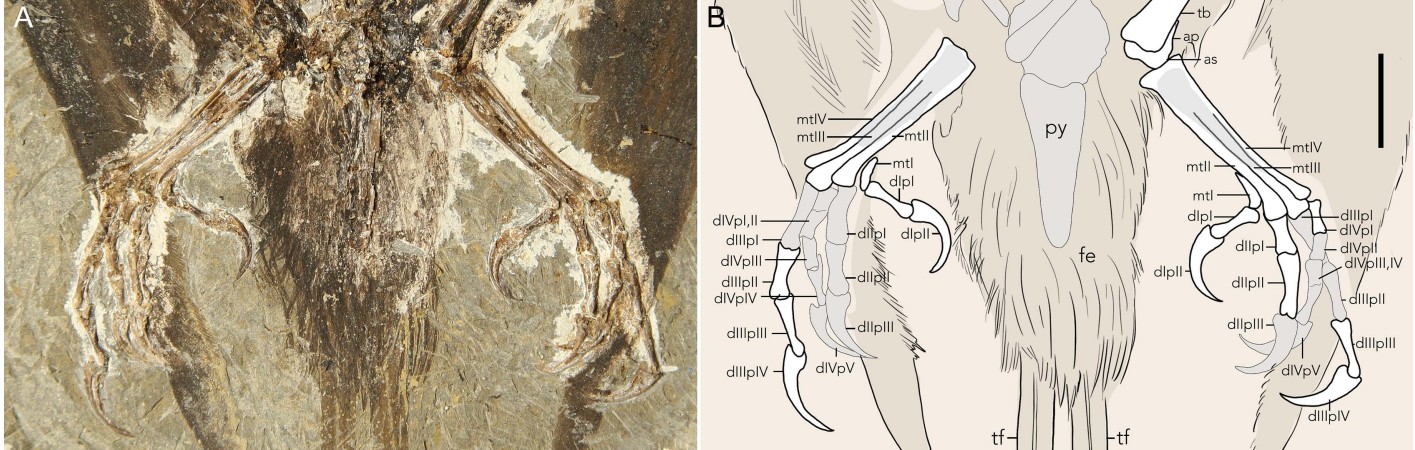

**Fig 4. The distal hindlimbs of STM11−4 (*Plumadraco bankoorum*).** (A) photo of the lower hindlimbs and caudal portion of the axial column (b) a line drawing. Well preserved bones are colored white, crushed or poorly preserved bones (or portions of bones) are grey, and preserved soft tissues are brown. Abbreviations: ap, ascending process; as, astragalus; d, digit; fe, feathers; p, phalanx; mt, metatarsal; p, phalanx; py, pygostyle; tb, tibia; tf, tail feather. Scale bar in equals 10 mm.

With the exception of the first phalanges of digit II and III, and phalanx 1 (and possibly 2) of digit IV, all other phalanges are shorter than the phalanx that immediately follows, resembling the condition in arboreal neornithines in which the individual phalanges lengthen distally within each digit [46]. Typical of enantiornithines, pedal digit III is the longest (pedal digit II and IV, 66% and 77% respectively the length of III). Phalanx 1 of pedal digit III is 122% the length of phalanx 2, contrasting with other enantiornithines in which phalanx 1 is shorter than phalanx 2 in pedal digit III (e.g., *Shangyang*, ~98%, *Yuanchuavis*, ~98%, *Zhouornis*, ~95%, and *Brevirostruavis*, ~95%). Measurements of pedal ungual curvature follows [45]. For pedal digits I – IV, the unguals measure approximately 101°, 131°, 134°, and 118° respectively. The ungual claws are less recurved than in other bohaiornithids (e.g., *Parabohaiornis*, *Bohaiornis*, and *Gretcheniao*) which have angles ranging from 93° - 126° [15,22].

**Soft tissues.** Body contour feathers, ranging in length from ~11–25 mm, radiate outwards from the skeleton beginning near the articulation of the frontal processes and nasals, extending around the head, neck, shoulders, chest, tibiae, proximal 1/4 of the tarsometatarsi, and the pygostyle. A clear outline of short crural feathers extends from the proximal region of the tibia and terminates 1/4–1/3 of the way down the tarsometatarsus. Contour feathers surround the pygostyle, and appear longer than those on other parts of the body, measuring ~18–25 mm compared to those of the head, neck, and shoulders which range from ~11–17 mm.

The wings are preserved folded, obscuring the number of flight feathers (remiges). The alula, best preserved on the right, originates along the proximocranial margin of the alular digit, measures 38.4 mm in length, and has a vane asymmetry of 4.4 [47,48]. On the right wing, the outer two primaries are less than half the length of the remaining primaries; they are well preserved with visible barbs. The longest primary of the right wing, preserving the complete leading and trailing vane widths, has an asymmetry ratio of 4.0, similar to values reported for extant volant birds (e.g., *Colius* 4.14, *Trogon* 4.18, *Phoeniconaias* 3.94) and other enantiornithines (e.g., *Eopengornis* 4.13, *Cruralispennia* 4.5) [48]. The longest primary feathers measure 105 and 100 mm on the left and right wings respectively. The shortest discernable primaries measure approximately 74 and 72 mm on the left and right wings respectively.

Two elongate racketed RDFs project straight distally out from the body contour feathers which surround the pygostyle, likely attaching to the laterodistal region of the pygostyle via soft tissue *in vivo* (Fig 1). The total length of

these feathers (293 mm) measures nearly twice that of the body (149 mm) (as measured by the total rostrocaudal length of the skull and the total length of the axial column) (Fig 5). Other enantiornithines with complete tail feathers have RDFs ranging from 1.0–1.6 times their body length. The rachis is comprised of three structures: the medial stripe, ramus, and lateral stripes (Figs 5,6B,C). These structures are also visible in other enantiornithine RDFs (e.g., *Orienantius, Dapingfangornis,* GSGM-07-CM-001) [40,8,41]. In contrast to the straight rachis, the outer-most margins, referred to as the ribbon-like sheet, have subtle undulating margins throughout their lengths, a feature also present in *Orienantius* [40]. The medial stripe begins to taper approximately 50% down the length of each RDF. Continuing distally, 75% down the length of each RDF, the lateral stripes terminate, and shortly after, the outer ribbon-like sheet structures abruptly transition and differentiate into pennaceous barbs forming small rackets (Fig 4C). These narrow, sharply-tapered rackets measure 8% the length of the feather compared to 10–54% in other enantiornithines bearing racketed RDFs [5,40,8,49,50]. The medial stripe and ramus mediolaterally taper and completely cease approximately halfway into the racket, similar to *Dapingfangornis,* whereas in *Orienantius* and GSGM-07-CM-001 they extend to the distal end of the feather [40,8,41]. The remainder of the racket's length consist of long pennaceous barbs extending from the distal end of the rachis (Figs 5,6C). The symmetrical, distally-tapered ellipse-shaped rackets of *Plumadraco* differ from the wider, distally rounded rackets in *Dapingfangornis* and the symmetrical, narrow oval rackets in *Orienantius* [40,41].

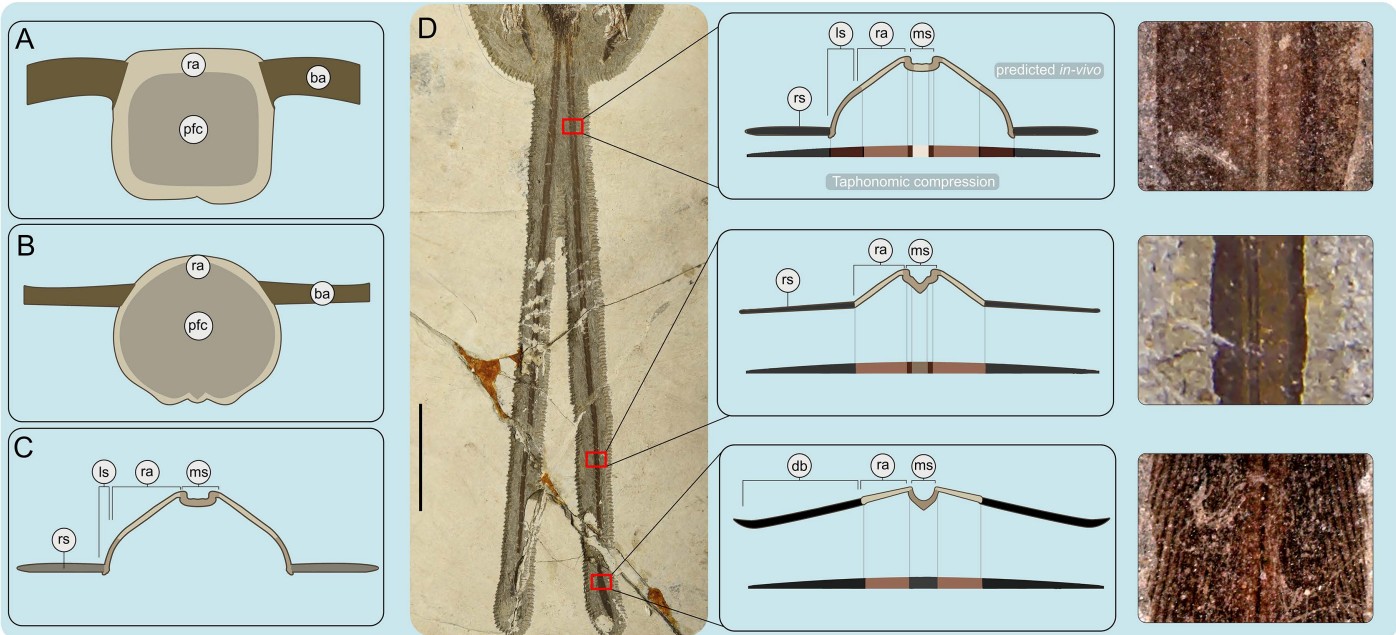

**Fig 5. Rectrix feather structures** The cross-sectional structure of the proximal portion of a rectrix in (A) *Antigone canadensis* (Gruiformes, Gruidae) (WWH-16868), and a tail covert (tail fan) in (B) *Pavo cristatus* (Galliformes, Phasianidae), compared with that (C) of the known three-dimensional structure of enantiornithine RDFs [modified from [10]. (D) The inferred three-dimensional shape of *Plumadraconis'* RDFs which changes down its length based on structural and color variations. Note how different structural dorsoventral thickness upon compression may affect color in a lithic specimen. Each sampled section down the length of the RDF is accompanied by a close-up photo of that portion of the feather to show differences in component color and width. Scale bar equals 50 mm. Abbreviations: ba, barbs; db, differentiated barbs; ls, lateral stripe; ms, medial stripe; pfc, pith-filled center; ra, ramus; rs, ribbon-like sheet.

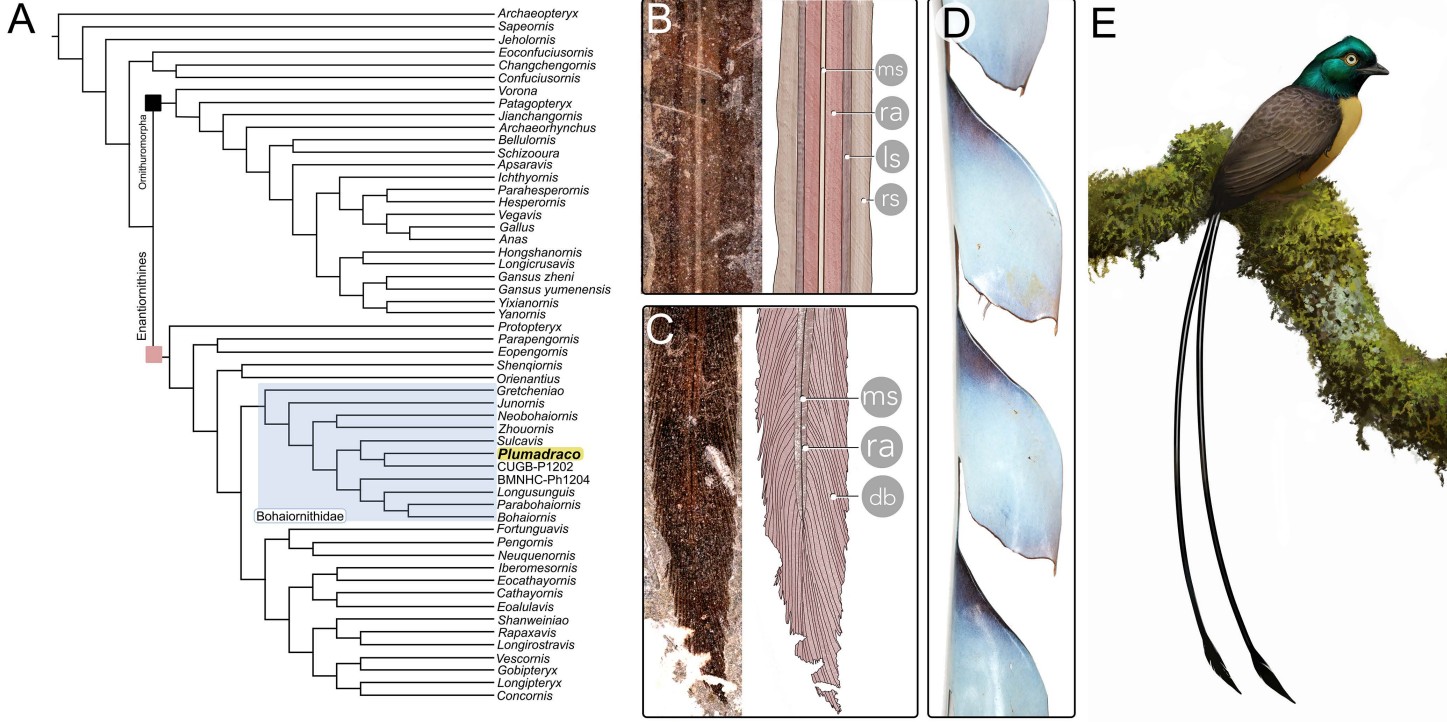

**Fig 6. Phylogenetic results and examples of exceptional preservation of RDFs. A)** Results of phylogenetic analyses suggest *Plumadraco* belongs to the diverse Bohaiornithidae. **B)** Finely-preserved structures present in the proximal portion of the right RDF of *Plumadraco*, C) and the preserved racket showing the differentiated barbs and the reduction, and eventual complete termination, of the central support structures (i.e., ramus and medial stripe). **(D)** The unique, undifferentiated barbs of the occipital plumes of *Pteridophora alberti* (King-of-Saxony Bird-of-Paradise) (FMNH 280831) forming tab-like laminated sheets. These structures may be uniquely analogous to the lateral margins (ribbon-like sheets) of enantiornithine RDFs preceding the distal ornaments. **(E)** An in-life restoration of *Plumadraco*. Illustration by Ville Sinkkonen.

## X-ray fluorescence (XRF) results

XRF analyses indicate that the preserved feathers differ from the matrix in having significantly lower relative quantities of Sr, Rb, Th, Zn, V, Ti, K, Al, P, and Si (S1 Table). PCA analysis resulted in feather and matrix data plotting in distinct groups with PC1 explaining 56.8%, PC2 16%, and PC3 11% of the variation (S1 Fig). XRF results show all feathers are proportionally high in CU and S compared to the matrix. These two elements are associated with the presence of darker color-producing melanin pigments [51]. The wings and rump feathers exhibit greater CU and S concentrations than the tail and head feathers. Evidence for pheomelanin pigments (responsible for red-brown colors), signaled when Zn is paired with similarly high levels of S [51], is not detected.

## Phylogenetic results

We conducted a heuristic search retaining the single shortest tree out of every 1000 trees followed by a second round of tree-bisection reconnection (TBR). The first round of TBR produced 1 tree with a best length of 955 steps; the second round of TBR produced 3 trees with a best length of 955 steps. The resulting phylogenetic tree placed *Plumadraco* within Enantiornithes, nested within bohaiornithids, specifically closely associated with CUGBP P1202 and *Sulcavis*. (Fig. 6A) [40,20]. Our results support the exclusion of *Shenqiornis* from Bohaiornithidae as previously suggested by [32] and [13] (See supplementary information for scoring). Inclusion within bohaiornithids according to the phylogenetic analyses is

supported by the width of the sternal end of the coracoid being at least half the length of the shaft [Character 70], the distal end of the femur with a contiguous ectocondylar tubercle and lateral condyle but without being developing a tibiofibular crest [Character 173], and trochlea III of the tarsometatarsus distally extending beyond that of II and IV [Character 201].

## Discussion

*Plumadraco* phylogenetically nests within Bohaiornithidae, a diverse group of Early Cretaceous mid-sized enantiornithines with previously documented, although poorly preserved, RDFs [12,13] (Fig 4A). Like all other Early Cretaceous enantiornithine, bohaiornithids are inferred to be arboreal based on hindlimb morphology, with the paleoenvironment of the Jehol Biota interpreted as a complex of lakes surrounded by forested wetlands with a seasonal, temperate climate [1,2,33,40,15]. The RDFs of STM11–4 are the best-preserved examples of this extinct feather morphotype among lithic enantiornithines. RDFs do not form a cohesive aerodynamic surface and would incur drag, strongly suggesting that they were either sexually-selected ornaments or for non-sexual conspecific communication [52,53]. Based on previous comparisons with extant birds and their variable presence in individual fossil specimens, particularly among those referable to the basal pygostylian *Confuciusornis sanctus*, RDFs are currently widely regarded as sexually-dimorphic ornaments [10,8,54]. In Early Cretaceous birds for which multiple soft-tissue preserving specimens are known (e.g., *Protopteryx*, *Confuciusornis*), RDFs are present in ~40–60% of individuals, a percentage consistent with sexual dimorphism [15,55-57]. The fine preservation of the RDFs in this specimen reveal new morphological details and previously unrecognized variation that in turn sheds light on their possible tensile properties and varying degrees of movement throughout their lengths. Morphological differences, both within the feathers themselves and when compared to other enantiornithines with similarly racketed RDFs, support interpretations these feathers were used in display behaviors [17].

### Detailed rectricial preservation reveals novel morphologies

The RDFs in STM11−4 are exceptionally well-preserved, revealing previously unrecognized morphological details, which contributes to the known structural diversity of these feathers (S2 Fig). Unlike all other enantiornithine specimens preserving RDFs, in which either the proximal or distalmost portions of the feathers are missing and/or poorly preserved, STM11−4 preserves the complete right RDF. The most noticeable attribute of *Plumadraco's* RDFs is their length, measuring 1.97 times that of the body (Fig 1). Previously, the proportionally longest RDFs were that of *Junornis*, measuring approximately 1.6 times that of the body [58] (S2 Table). Artificially exaggerating tail plumage length in males of sexually dimorphic extant birds leads to increased breeding success and frequency (e.g., *Parotia, Euplectes, Vidua, Hirundo, Gallus, Panurus*) [59-64 65] indicating that costly features, such as proportionally elongate rectrices, are targeted by female selection [59,65,66,64]. This has led to the suggestion that elongate ornamental feathers in Mesozoic birds evolved under similar pressures [53,9].

For much of the RDF's length (~85%) in STM11−4, the lateral margins are lined by narrow ribbon-like structures [8]. These structures differ from the linear rachis in having subtly undulating outer margins. This is not visible in most other poorly preserved specimens although reinspection reveals that this morphology can also be observed in *Orienantius* [15,12,40] (Figs 1, 6B). These differences in observable structure suggest different degrees of pliability between the rachis and ribbon-like structures, providing additional support for the interpretation that this ribbon-like structure consists of short, undifferentiated barbs, in which the normal inter-barb cell death that results in individualized feather barbs, fails to occur [8]. Similar processes of either barb fusion or lack of cell death during development are responsible for unusual feather structures in neornithines, such as the enamel-like tabbed occipital plumes of *Pteridophora alberti* (King of Saxony bird-of-paradise) [67,68] (Fig 6D).

Enantiornithine RDF racket-plumes are widespread across the clade (e.g., bohaiornithids, *Protopteryx, Paraprotopteryx, Orienantius, Dapingfangornis,* 'Cratoavis', Kachin amber specimens), [15,40,22,9,50] and both racket

proportions and shape vary considerably between taxa. Similar variation is observed among extant neornithines whose rackets function to draw conspecific visual attention, particularly in forested environments, as they contrast in shape with the often-vertical background vegetation both when perched and while volant [69,68]. Notably, most enantiornithines including all taxa documented as having racket-plumes are widely hypothesized to occupy arboreal environments [1,53]. This suggests that the rackets in *Plumadraco* and other enantiornithines may have similarly produced conspicuous signals utilized in intraspecific communication or courtship displays like many forest-dwelling birds today (e.g., momotids, trochilids, paradisaeids, alcedinids) [70,71,68,72].

The fine preservation of the RDFs in STM11−4 reveal previously undocumented variation both within the racket itself and between other enantiornithines that supports this interpretation. In *Plumadraco*, the rachis diminishes halfway down the racket and completely ceases soon after. Reduction of the rachis weakens the distal portion of the ornament, a trait referred to as enfeeblement (Fig 6). Enfeeblement of the distal portion of proportionally elongate feathers exaggerates their visual signal by allowing greater degree of flexion of the ornament (e.g., racket) relative to the proximal, comparatively more rigid, portions of the same feather when the rectrices are moved, often resulting in "flickering" motions. This is exemplified by the ornamental rump coverts of the extant *Pavo* [73] (Fig 7B). Similar distal enfeeblement is also observed in *Dapingfangornis* [41], but contrasts with the condition in some other enantiornithines where the rachis extends farther into (e.g., *Orienantius*), or completely through (e.g., GSGM-07-CM-001) each racket, indicating a lack of enfeeblement and greater distal rigidity compared to *Plumadraco* and *Dapingfangornis* [40,8] (Fig 7, S3 Fig). These differences in enfeeblement between enantiornithines suggests interspecific variation in the movement of the distal racket when the tail feathers are raised or depressed. This variation is consistent with their preservation as well. Both *Plumadraco* and *Dapingfangornis* preserve the enfeebled portions of the rackets angling away from the main axis of the feather, whereas those with distally extending (non-enfeebled) rachises preserve the racket aligned with the proximal rachis (S3 Fig).

This interpretation is also supported by inferences regarding enantiornithine tail musculature, which has been suggested to be indicative of dorsoventral tail movements of the RDFs during displays [17,53]. Soft tissue traces associated with the enantiornithine pygostyle indicate a reduction (or complete absence) of the *mm. bulbi rectricium* and *lateralis caudae*, which suggests a limited ability to mediolaterally wag or spread the rectrices [24,53,74]. In contrast, enantiornithine

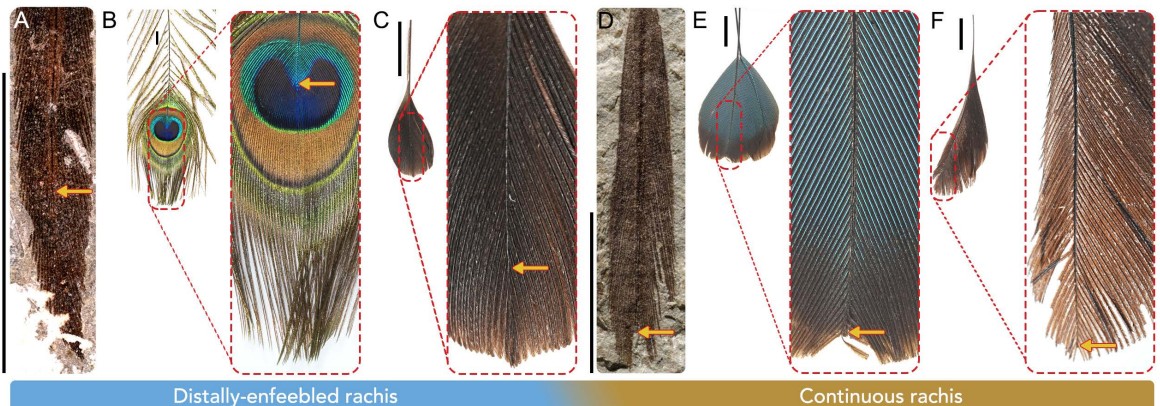

**Fig 7. Subtle structural differences in distal portions of rectrices.** A-C, examples of rectrices (rump covert in *Pavo*) which show distal enfeeblement (exaggerated tapering or termination) of the rachis within ornaments, while D-F shows examples of rectrices with the rachis more continuous throughout the distal ornaments. Arrows indicate where in the ornament the rachis terminates. **A)** *Plumadraco*, **B)** *Pavo cristatus* (FMNH 21426, sexually-dimorphic), **C)** *Prioniturus discurus* (FMNH 11554, sexually-dimorphic), **D)** *Orienantius* ([40]; Elsevier, *Cretaceous Research*), **E),** *Momotus motmota* (FMNH 299790, monomorphic), **F)** *Dicrurus paradiseus* (FMNH 212154, monomorphic). All scale bars equal 10 mm.

pygostyles have well-developed insertion surfaces for the *mm. levator* and *depressor caudae*, which raise and depress (i.e., pump) the tail, respectively [24,75,53]. Dorsoventrally pumping or sustained dorsiflexion of elongate, distally-ornamented rectrices is a behavior often seen in the courtship displays of sexually-dimorphic species (e.g., *Argus, Centrocercus, Cicinnurus, Diphyllodees, Loddigesia, Ocreatus, Paradisaea*) [75,76,68,65,77]. This type of tail movement has previously been suggested for at least one enantiornithine with an extravagant tail morphology consisting of three distinct feather morphotypes (e.g., *Feitianius*) [17,53]. Distal enfeeblement of the racket in some enantiornithines would increase the overall visual effect of such a courtship display. Overall, these observable variations in RDF length, racket size, and racket shape (e.g., ellipse, oval, circular) among enantiornithines, suggest displays unique to each taxon.

## Support for sexual dimorphism based on known enantiornithine nesting strategies

Enantiornithine growth strategies differ from all modern birds in that they hatched highly precocial (e.g., early development of the forelimb), required several years to achieve adult size, and grew their RDFs before both sexual and somatic maturity [78,55,79]. These inherent differences in growth and development between neornithines and enantiornithines make it difficult to determine the evolutionary drivers that produced RDFs. Despite these tremendous life history differences, some environmental pressures can be expected to be similar in birds across time, such as mitigating predation risks during nesting.

Available evidence of enantiornithine nesting strategies are both geographically and geologically limited, having been described from 86, 72, and 70 My Romanian, Mongolian, and Argentinean deposits respectively [80-82]. These sites are found in terrestrial deposits that record colonial open ground nests, in which single eggs or small clutches of eggs were embedded in sediment [80,82]. Egg remains and embryonic bones are associated with adult remains at the Oarda de Jos locality in Romania, tentatively suggesting some form of parental care (i.e., nest guarding, possible contact incubation) during embryonic development [80]. The microstructure of the cuticle preserved on an intra-abdominal egg in *Avimaia* IVPP V25371 (~ 120 My) provides indirect evidence that Early Cretaceous enantiornithines also nested on the ground in humid, damp riparian environments [83]. Although data is limited, ground nests are only likely to preserve in terrestrial deposits and almost all Early Cretaceous enantiornithines are known from aquatic deposits. Ground nesting is documented in non-avian paravians and considered plesiomorphic to Aves and Enantiornithes and was evidently retained during enantiornithine evolution into the Late Cretaceous [81,84]. Though the ecologies of the enantiornithine taxa for which eggs and nests have been recovered may have differed from that of *Plumadraco*, the inferred nesting strategy similarities across all of them suggests a conserved, and likely widespread behavior present in most enantiornithines.

Compared to cavity nesters, tending to open-ground nests increases the adult's exposure to potential predators. In extant sexually dimorphic bird species with open nests, though the male may be ornamented, females, which more often brood and tend to the nest, typically have cryptic plumages to reduce the risk of predation to both themselves and their young [69,85-90]. The need for crypsis often leads to proportionally reduced rectrix lengths, reduced hue saturation, and increased camouflage patterns which help the female remain inconspicuous during nest attendance [69,85,86,87,89]. In contrast, cavity-nesting birds can forgo crypsis as concealment alone lowers predation risks, and indeed, nearly all elongate-tailed monomorphic species are cavity nesters and produce highly altricial young (e.g., coraciiforms, psittaciforms) [91,69,92,93,77,94,95]. Precocial offspring, like in all known enantiornithines, often emancipates males from rearing duties, with costly ornamental plumages in males often associated with proportionally limited parental care compared to females [65]. The ground nesting behavior and precocial developmental strategy utilized by enantiornithines therefore supports the hypothesis that RDFs are sexually dimorphic. Female enantiornithines, which likely tended to ground nests, would have benefitted from inconspicuous plumage (e.g., reduced rectrix length) and coloration.

While enantiornithine biology may be informed by extant birds [96-98], no direct analogues for these birds exist, limiting our ability to predict the primary evolutionary drivers of RDFs. Based on available evidence, we suggest STM11−4 represents a male individual whose RDF morphology and length were utilized in courtship displays and evolved as the result

of targeted conspecific female selection. However, direct evidence of enantiornithine nesting strategies during the Early Cretaceous and additional information concerning enantiornithine tail musculature are needed to further corroborate this interpretation.

## Supporting information

**S1 Table. XRF data from specimen STM11−4.**
(XLSX)

**S2 Table. Known enantiornithine RDFs with preserved features.**
(XLSX)

**S1 Fig. PCA results of XRF data from specimen STM11–4.** Feathers and matrix exhibit different chemical signatures, supporting the true presence of both the body covering and the elongate tail feathers. PC1–3 represent abundances of significant elements (detailed in Table S1). Inset photos of the specimen show where samples were taken. Abbreviations: f, feather; m, matrix.
(PNG)

**S2 Fig. Published enantiornithine specimens with evidence of RDFs Streamers–** (A) Parapengornis [34], (B) *Eopengornis* [99], (C) indet. Enantiornithine GSGM-07-CM-001; Rackets – (D) *Bohaiornis* [12], (E) *Dapingfangornis* [5], (F) *Junornis* [58], (G) *Orienantius* [40], (H) *Misuvavis* [42], (I) *Protopteryx* [57], (J) *Paraprotopteryx* [50], (K) *Shanweiniao* [14], (L) Enantiornithine indet. IVPP V 15564 [55], (M) Enantiornithine indet. STM 34−9 [55], (N) Bohaiornithid indet. CUGB P1202 [23]; Fanned or multiple morphotypes – (O) *Chiappeavis* [6], (P) *Feitianius* [100] , (Q) *Yuanchuavis* [101] .
(PNG)

**S3 Fig. Predicted differences in racket movement reflected in specimens.** Proximally to distally, the red points indicate (1) the start of the racket, (2) the termination of the rachis, and (3) the distal margin of the racket. Yellow lines measure the angle using the pre-racket rachis along the midline, the start of the racket, and the racket's distal central margin. Angles were measured using ImageJ. (A) *Plumadraco*, (B) *Dapingfangornis* (C) *Orienantius*, and (D) CAGS-IG-07-CM-001. Note the proportional differences between points 2 and 3 among sampled taxa with enfeeblement occurring within (A) and (B) more so than (C) and (D). A greater distance between these points would facilitate greater movement of the intermediate space, as made evident by angular differences among *Plumadraco* and *Dapingfangornis* compared to *Orienantius* and CAGS-IG-07-CM-001 (which show no discernable angular difference).
(PNG)

## Acknowledgments

We thank Xiaomei Zhang, Xuwei Yin, and Shiying Yin for facilitating our visit to the STM. We are grateful to Laure Dussubieux (FMNH Elemental Analysis Facility) for us of the XRF and Ben Marks and Dave Willard for access to bird collections at the FMNH. We thank Ville Sinkkonen for the illustration of *Plumadraco* used in Fig 6 and Dr. Trevor Price of the University of Chicago for advice and guidance on the content included in this manuscript.

## Author contributions

**Conceptualization:** Alexander D. Clark, Jingmai K. O'Connor, Xiaoli Wang, Yan Wang, Stephen Pruett-Jones, Xiangyu Zhang, Xing Wang, Xiaoting Zheng, Zhonghe Zhou.

**Data curation:** Alexander D. Clark.

**Formal analysis:** Alexander D. Clark.

**Investigation:** Alexander D. Clark, Jingmai K. O'Connor.

**Methodology:** Alexander D. Clark, Jingmai K. O'Connor.

**Visualization:** Alexander D. Clark.

**Writing – original draft:** Alexander D. Clark.

**Writing – review & editing:** Jingmai K. O'Connor, Stephen Pruett-Jones.

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
