## [Decision Letter · Decision Letter 0]

11 Mar 2026

PONE-D-26-04011Hyperelongate ornamental tail feathers in a new Early Cretaceous enantiornithine bird PLOS One

Dear Dr. Clark,

Thank you for submitting your manuscript to PLOS ONE. After careful consideration, we feel that it has merit but does not fully meet PLOS ONE’s publication criteria as it currently stands. Therefore, we invite you to submit a revised version of the manuscript that addresses the points raised during the review process.

We look forward to receiving your revised manuscript.

Kind regards,

Jun Liu

Academic Editor

PLOS One

Journal Requirements:

2.  Please take this opportunity to be sure you have met all of our guidelines for new species. For proper registration of a new zoological taxon, we require two specific statements to be included in your manuscript.

A.         In the Results section, the globally unique identifier (GUID), currently in the form of a Life Science Identifier (LSID), should be listed under the new species name, for example:

Anochetus boltoni Fisher sp. nov. urn:lsid:zoobank.org:act:B6C072CF-1CA6-40C7-8396-534E91EF7FBB

Another LSID for the manuscript itself should also appear within the Nomenclature statement. You will need to contact Zoobank (zoobank.org/About) to obtain a GUID (LSID). You should receive one LSID for your manuscript and a separate, unique LSID for the new species.

B.         Please also insert the following text into the Methods section, in a sub-section to be called ""Nomenclatural Acts"":

The electronic edition of this article conforms to the requirements of the amended International Code of Zoological Nomenclature, and hence the new names contained herein are available under that Code from the electronic edition of this article. This published work and the nomenclatural acts it contains have been registered in ZooBank, the online registration system for the ICZN. The ZooBank LSIDs (Life Science Identifiers) can be resolved and the associated information viewed through any standard web browser by appending the LSID to the prefix ""http://zoobank.org/"". The LSID for this publication is: urn:lsid:zoobank.org:pub: XXXXXXX. The electronic edition of this work was published in a journal with an ISSN, and has been archived and is available from the following digital repositories: PubMed Central, LOCKSS [author to insert any additional repositories].

All PLOS ONE articles are deposited in PubMed Central and LOCKSS. If your institute, or those of your co-authors, has its own repository, we recommend that you also deposit the published online article there and include the name in your article.

Following a recent ruling by the International Commission on Zoological Nomenclature, electronic journals are now a valid format for publication of new zoological taxa. In order to ensure the valid publication of your new species, please be sure to include the updated version of Nomenclatural Acts (above). A complete explanation of our guidelines for publishing new species can be found on our website: http://www.plosone.org/static/guidelines#zoological.

3. Please note that PLOS One has specific guidelines on code sharing for submissions in which author-generated code underpins the findings in the manuscript. In these cases, we expect all author-generated code to be made available without restrictions upon publication of the work. Please review our guidelines at https://journals.plos.org/plosone/s/materials-and-software-sharing#loc-sharing-code and ensure that your code is shared in a way that follows best practice and facilitates reproducibility and reuse.

4. In your manuscript, please provide additional information regarding the specimens used in your study. Ensure that you have reported human remain specimen numbers and complete repository information, including museum name and geographic location.

For more information on PLOS One's requirements for paleontology and archeology research, see https://journals.plos.org/plosone/s/submission-guidelines#loc-paleontology-and-archaeology-research.

5. Thank you for stating in your Funding Statement:

“This research was in part funded by the Taishan Scholar Foundation of Shandong Province (Ts20190954) and the National Natural Science Foundation of China (NSFC), grant numbers (42288201, 42572027).”

6. Thank you for stating the following financial disclosure:

“This research was in part funded by the Taishan Scholar Foundation of Shandong Province (Ts20190954) and the National Natural Science Foundation of China (NSFC), grant numbers (42288201, 42572027).”

7. Please amend the manuscript submission data (via Edit Submission) to include author Dr. Yan Wang.

8. We note that Figure 6 and 7 in your submission contain copyrighted images. All PLOS content is published under the Creative Commons Attribution License (CC BY 4.0), which means that the manuscript, images, and Supporting Information files will be freely available online, and any third party is permitted to access, download, copy, distribute, and use these materials in any way, even commercially, with proper attribution. For more information, see our copyright guidelines: http://journals.plos.org/plosone/s/licenses-and-copyright.

1. You may seek permission from the original copyright holder of Figure 6 and 7 to publish the content specifically under the CC BY 4.0 license.

**Additional Editor Comments:**

Please revise ms following the reviewer's comments, especilly the places need to be clarified.

Reviewers' comments:

Reviewer's Responses to Questions

**Comments to the Author**

1. Is the manuscript technically sound, and do the data support the conclusions?

Reviewer #1: Yes

Reviewer #2: Yes

2. Has the statistical analysis been performed appropriately and rigorously? 

Reviewer #1: N/A

Reviewer #2: N/A

3. Have the authors made all data underlying the findings in their manuscript fully available?

Reviewer #1: Yes

Reviewer #2: Yes

4. Is the manuscript presented in an intelligible fashion and written in standard English?

Reviewer #1: Yes

Reviewer #2: Yes

5. Review Comments to the Author

Reviewer #1: General Assessment

This manuscript represents a significant contribution to avian paleontology. The identification and detailed description of Plumadraco bankoorum and its unique Rachis-Dominated Feathers (RDFs) provide compelling evidence for complex sexual selection pressures in Early Cretaceous birds. Addressing the suggested revisions, particularly those concerning methodological clarity, nuanced interpretations of data, and consistent terminology, will further strengthen the manuscript and enhance its impact. I anticipate that these revisions will lead to an even more robust and influential publication.

Specific Recommendations for Clarification

Nesting Ecology and Morphology Correlations:

Clarification is needed regarding the inferences linking nesting ecology to tail morphology. Correlations should be treated with caution, as the specific ecology of this new taxon may differ from other fossils used for comparison. Please ensure that assumptions about nesting behavior do not overextend the available evidence.

Definition of Rachis-Dominated Feathers (RDFs):

The morphological definition of RDFs requires further illustration. Typically, "rachis-dominated" implies that the rachis comprises the major part of the feather structure; however, barbs are clearly observed in this specimen (particularly within the rackets). Please reconcile this terminology with the observed morphology to avoid confusion.

Phylogenetic Support Values:

Please provide support indices (e.g., bootstrap or jackknife values) for the TNT phylogenetic analysis. Including these statistical values will strengthen the robustness of the cladistic results.

Pygostyle Morphology and Display Mechanics:

Fan-shaped tail feathers and rectrices are generally understood to have evolved in Neornithines, often associated with a short pygostyle. However, in Enantiornithines, which possess an elongate pygostyle, the mechanical effect of this structure on tail feather display remains unclear. Please discuss how the long, shaped pygostyle might influence tail feather display mechanics compared to neornithines.

Reviewer #2: **Comments to the Author(s)**

This study describes a new enantiornithine bird and reports an unusual hyperelongate ornamental tail feather among early birds. Overall, this manuscript is well-written and the points are quite straightforward. I only have a few suggestions and recommendations that I believe might improve the quality of this manuscript:

1) Lines 21–22: In the abstract, I am not sure whether this comparison is appropriate — passerines today are considerably more diverse, clearly far exceeding the diversity level seen in enantiornithines.

2) Line 115: It is not very common to place the measurement table immediately here. Consider moving it to the location where measurements are first mentioned in the text.

3) Line 123: It would be more appropriate to refer to the holotype rather than simply the "fossil specimen" of Plumadraco.

4) Lines 145–154: In the Diagnosis section, it would be better to separate the characters that support referral to Enantiornithes from those that further support assignment to Bohaiornithidae. Additionally, many of the features used to assign this taxon to bohaiornithids — such as the furcula, coracoid, sternum, and sacral vertebrae — are indicated as crushed in the line drawing figure.

5) Line 153: Please indicate that this refers specifically to the lateral trabeculae of the sternum.

6) Lines 209–210: The rami in this new specimen are slightly bowed. Could this bowing account for the apparently narrow interclavicular angle, or might there be inconsistency in how this angle is measured across taxa?

7) Line 214: It is unclear whether "pterygoma" is the appropriate term for the coracoid body.

8) Lines 230–232: To me, the left trabecula appears more caudolaterally oriented rather than purely caudally oriented.

9) Line 266: Since the ungual phalanx of the minor digit is commonly very tiny and easily lost during preservation, it may be better not to place too much emphasis on the phalangeal formula being 2-3-1 here.

10) Lines 370–371: It is unclear how a precise 3D structure can be reconstructed from 2D observations alone.

11) Line 402: This should read "exceptional preservation of RDFs".

12) Lines 406–407: In Figure 6D, it would be helpful to indicate the exact location of the barbs within the tail feather of the modern bird-of-paradise.

13) Lines 426–429: The use of percentage values may be appropriate for Confuciusornis, given the large number of available specimens, but may not be applicable to enantiornithines, for which most taxa are represented by very few specimens.

14) Lines 453–454: The authors discuss differences in RDF morphology between STM11-4 and other fossil birds, yet only Orienantius is included in the comparison shown in Figure 7. It would be helpful to include additional enantiornithine tail feathers in this figure for a more comprehensive comparison.

6. PLOS authors have the option to publish the peer review history of their article (what does this mean?). If published, this will include your full peer review and any attached files.

Reviewer #1: No

Reviewer #2: No

---

## [Author Response · Author response to Decision Letter 1]

24 Mar 2026

Editorial comments requiring responses

Please take this opportunity to be sure you have met all of our guidelines for new species. For proper registration of a new zoological taxon, we require two specific statements to be included in your manuscript.

We have included the required LSID values for the new taxon underneath the introduction of its new name

We note that Figure 6 and 7 in your submission contain copyrighted images. All PLOS content is published under the Creative Commons Attribution License (CC BY 4.0), which means that the manuscript, images, and Supporting Information files will be freely available online, and any third party is permitted to access, download, copy, distribute, and use these materials in any way, even commercially, with proper attribution. For more information, see our copyright guidelines: http://journals.plos.org/plosone/s/licenses-and-copyright.

Figure 6E is the only illustration created by someone who was not an author. Permissions have been secured and the document attached as directed.

Reviewer #1: General Assessment

Specific Recommendations for Clarification

Nesting Ecology and Morphology Correlations:

Clarification is needed regarding the inferences linking nesting ecology to tail morphology. Correlations should be treated with caution, as the specific ecology of this new taxon may differ from other fossils used for comparison. Please ensure that assumptions about nesting behavior do not overextend the available evidence.

Thank you very much for this suggestion. We have made sure to not overextend our interpretations drawn from other enantiornithines – only that, based on available evidence, our conclusions suggest a shared ground-nesting strategy across enantiornithines. We outline the available evidence from three different enantiornithines across three different geological times to suggest that this behavior is likely shared across the group – based on all available nesting evidence within this group. We then discuss the behavioral implications of this nesting strategy in modern birds and how this would affect enantiornithine behavior and probably parental roles (and likely in all birds across time). We have made sure to edit the text to also reflect the uncertainty that the reviewer has pointed out. Thank you!

Definition of Rachis-Dominated Feathers (RDFs):

The morphological definition of RDFs requires further illustration. Typically, "rachis-dominated" implies that the rachis comprises the major part of the feather structure; however, barbs are clearly observed in this specimen (particularly within the rackets). Please reconcile this terminology with the observed morphology to avoid confusion.

We have modified the definition of “RDF” in our introduction to state that they are “feathers with proportionately wide rachi throughout majority of their lengths, referred to as rachis-dominated feathers”. By stating “majority”, we eliminate confusion that they maintain their wide width for the entire feather – which tends to never happen in any enantiornithine RDF with the full length preserved.

Phylogenetic Support Values:

Please provide support indices (e.g., bootstrap or jackknife values) for the TNT phylogenetic analysis. Including these statistical values will strengthen the robustness of the cladistic results.

Given that are results reflect those of a 50% majority tree, we find these values to be unnecessary.

Pygostyle Morphology and Display Mechanics:

Fan-shaped tail feathers and rectrices are generally understood to have evolved in Neornithines, often associated with a short pygostyle. However, in Enantiornithines, which possess an elongate pygostyle, the mechanical effect of this structure on tail feather display remains unclear. Please discuss how the long, shaped pygostyle might influence tail feather display mechanics compared to neornithines.

Thank you for the suggestion, and a great point to bring up! The full biomechanical effects of elongate enantiornithine pygostyles on rectricial flexion is a topic that cannot be adequately covered in this manuscript as it entails another major physics-based manuscript-length report. For example, a longer moment arm (longer pygostyle) results in faster flexion and more exaggerated speed the further you move from the fulcrum – but these results and discussion require research outside the scope of this manuscript. These pygostyles would have undoubtedly affected the mechanical advantage and moment arm mechanics across taxa (e.g., pengornithids – broad and short, compared to bohaiornithids – distally tapered and long).

In trying to avoid overextending the evidence we have, we have only discussed the plausible implications of pygostyle morphology/insertion surfaces in the discussion (543-550).

Reviewer #2: **Comments to the Author(s)**

This study describes a new enantiornithine bird and reports an unusual hyperelongate ornamental tail feather among early birds. Overall, this manuscript is well-written and the points are quite straightforward. I only have a few suggestions and recommendations that I believe might improve the quality of this manuscript:

1) Lines 21–22: In the abstract, I am not sure whether this comparison is appropriate — passerines today are considerably more diverse, clearly far exceeding the diversity level seen in enantiornithines.

Revised. Mention of passerines has been removed

2) Line 115: It is not very common to place the measurement table immediately here. Consider moving it to the location where measurements are first mentioned in the text.

Revised. Now moved into the results section

3) Line 123: It would be more appropriate to refer to the holotype rather than simply the "fossil specimen" of Plumadraco.

Revised

4) Lines 145–154: In the Diagnosis section, it would be better to separate the characters that support referral to Enantiornithes from those that further support assignment to Bohaiornithidae. Additionally, many of the features used to assign this taxon to bohaiornithids — such as the furcula, coracoid, sternum, and sacral vertebrae — are indicated as crushed in the line drawing figure.

Revised. The features that associate this individual with bohaiornithids are now clarified. Portions of the furcula, coracoid, sternum, and sacral vertebrae, while crushed, allow for enough morphological description that indicate both enantiornithine and specifically, bohaiornithid affinities.

5) Line 153: Please indicate that this refers specifically to the lateral trabeculae of the sternum.

Revised

6) Lines 209–210: The rami in this new specimen are slightly bowed. Could this bowing account for the apparently narrow interclavicular angle, or might there be inconsistency in how this angle is measured across taxa?

The angle is measured from the distal-most omal tips to the midline of the joined rami. The lateral bowing margins do not affect the inter-clavicular angle.

7) Line 214: It is unclear whether "pterygoma" is the appropriate term for the coracoid body.

Revised. “pterygoma” changed to “corpus”

8) Lines 230–232: To me, the left trabecula appears more caudolaterally oriented rather than purely caudally oriented.

Agreed – thank you for this suggestion. This section has been revised.

9) Line 266: Since the ungual phalanx of the minor digit is commonly very tiny and easily lost during preservation, it may be better not to place too much emphasis on the phalangeal formula being 2-3-1 here.

Revised and removed

10) Lines 370–371: It is unclear how a precise 3D structure can be reconstructed from 2D observations alone.

Three dimensional enantiornithine RDFs are used as a template to infer the homologous structure also likely present in Plumadraco.

11) Line 402: This should read "exceptional preservation of RDFs".

Revised

12) Lines 406–407: In Figure 6D, it would be helpful to indicate the exact location of the barbs within the tail feather of the modern bird-of-paradise.

Thank you for the suggestion. But in the manuscript and in the figure caption, we state that they are “occipital” plumes. These feathers actually appear as-in throughout their length, erupting from just behind the eye.

13) Lines 426–429: The use of percentage values may be appropriate for Confuciusornis, given the large number of available specimens, but may not be applicable to enantiornithines, for which most taxa are represented by very few specimens.

We very much agree, which is why we only do so for the only, currently published, enantiornithine with multiple specimens preserving soft tissues (Protopteryx).

14) Lines 453–454: The authors discuss differences in RDF morphology between STM11-4 and other fossil birds, yet only Orienantius is included in the comparison shown in Figure 7. It would be helpful to include additional enantiornithine tail feathers in this figure for a more comprehensive comparison.

Thank you for the suggestion. Other enantiornithines bearing elongate RDFs with their distal tips present are figured in Fig. S3. The primary objective of figure 7 is the comparison with extant birds to illustrate that there are in fact differences in distal rachis termination which results in varying levels of plasticity/rigidity.

---

## [Editor Report · Decision Letter 1]

6 Apr 2026

Hyperelongate ornamental tail feathers in a new Early Cretaceous enantiornithine bird

PONE-D-26-04011R1

Dear Dr. Clark,

We’re pleased to inform you that your manuscript has been judged scientifically suitable for publication and will be formally accepted for publication once it meets all outstanding technical requirements.

Kind regards,

Jun Liu

Academic Editor

PLOS One

Additional Editor Comments (optional):

Did the last author really read this version? Zhonghe's affiliation should be changed as only "Key Laboratory of Vertebrate Evolution and Human Origins, Institute of Vertebrate Paleontology and Paleoanthropology, Chinese Academy of Sciences, Beijing 100044, China"
---

## [Editor Report · Acceptance letter]

PONE-D-26-04011R1

PLOS One

Dear Dr. Clark,

I'm pleased to inform you that your manuscript has been deemed suitable for publication in PLOS One. Congratulations! Your manuscript is now being handed over to our production team.

Kind regards,

on behalf of

Dr. Jun Liu

Academic Editor

PLOS One